# Ribociclib Induces Broad Chemotherapy Resistance and EGFR Dependency in ESR1 Wildtype and Mutant Breast Cancer

**DOI:** 10.3390/cancers13246314

**Published:** 2021-12-16

**Authors:** Isabel Mayayo-Peralta, Beatrice Faggion, Liesbeth Hoekman, Ben Morris, Cor Lieftink, Isabella Goldsbrough, Lakjaya Buluwela, Joseph C. Siefert, Harm Post, Maarten Altelaar, Roderick Beijersbergen, Simak Ali, Wilbert Zwart, Stefan Prekovic

**Affiliations:** 1Division of Oncogenomics, Oncode Institute, The Netherlands Cancer Institute, 1066 CX Amsterdam, The Netherlands; i.mayayo@nki.nl (I.M.-P.); beatrice.faggion@novartis.com (B.F.); j.siefert@nki.nl (J.C.S.); 2Proteomics Facility, The Netherlands Cancer Institute, 1066 CX Amsterdam, The Netherlands; l.hoekman@nki.nl (L.H.); m.altelaar@nki.nl (M.A.); 3Division of Molecular Carcinogenesis and Robotics and Screening Centre, The Netherlands Cancer Institute, 1066 CX, Amsterdam, The Netherlands; b.morris@nki.nl (B.M.); c.lieftink@nki.nl (C.L.); r.beijersbergen@nki.nl (R.B.); 4Department of Surgery & Cancer, Imperial College London, London SW7 2BX, UK; isabella.goldsbrough16@imperial.ac.uk (I.G.); l.buluwela@imperial.ac.uk (L.B.); simak.ali@imperial.ac.uk (S.A.); 5Biomolecular Mass Spectrometry and Proteomics, Bijvoet Center for Biomolecular Research and Utrecht Institute for Pharmaceutical Sciences, Utrecht University, 3584 CS Utrecht, The Netherlands; h.post@uu.nl; 6Laboratory of Chemical Biology and Institute for Complex Molecular Systems, Department of Biomedical Engineering, Eindhoven University of Technology, 5612 AZ Eindhoven, The Netherlands

**Keywords:** oestrogen receptor, breast cancer, cdk4/6 inhibitors, chemotherapy resistance, EGFR signalling

## Abstract

**Simple Summary:**

Disease progression while receiving treatment is a major problem in breast cancer. Mutations in the oestrogen receptor-α often lead to loss of drug activity, resulting in an inability of anti-oestrogens to stop cancer growth. There is an urgent need to establish which therapies can effectively eradicate cancer cells and also to understand how these therapies work. The aim of this study was to find compounds that diminish viability of breast cancer cells harbouring mutated oestrogen receptor-α. We discovered that cells, regardless of their oestrogen receptor-α mutational status, are vulnerable to the cell-cycle inhibitor ribociclib, which causes senescence accompanied by a decrease in sensitivity to various chemotherapies. Importantly, we found that viability of ribociclib-induced senescent cells is maintained by the EGFR signalling pathway, which may be therapeutically exploited.

**Abstract:**

While endocrine therapy is highly effective for the treatment of oestrogen receptor-α (ERα)-positive breast cancer, a significant number of patients will eventually experience disease progression and develop treatment-resistant, metastatic cancer. The majority of resistant tumours remain dependent on ERα-action, with activating *ESR1* gene mutations occurring in 15–40% of advanced cancers. Therefore, there is an urgent need to discover novel effective therapies that can eradicate cancer cells with aberrant ERα and to understand the cellular response underlying their action. Here, we evaluate the response of MCF7-derived, CRISPR-Cas9-generated cell lines expressing mutant ERα (Y537S) to a large number of drugs. We report sensitivity to numerous clinically approved inhibitors, including CDK4/6 inhibitor ribociclib, which is a standard-of-care therapy in the treatment of metastatic ERα-positive breast cancer and currently under evaluation in the neoadjuvant setting. Ribociclib treatment induces senescence in both wildtype and mutant ERα breast cancer models and leads to a broad-range drug tolerance. Strikingly, viability of cells undergoing ribociclib-induced cellular senescence is maintained via engagement of EGFR signalling, which may be therapeutically exploited in both wildtype and mutant ERα-positive breast cancer. Our study highlights a wide-spread reduction in sensitivity to anti-cancer drugs accompanied with an acquired vulnerability to EGFR inhibitors following CDK4/6 inhibitor treatment.

## 1. Introduction

Breast cancer is the most commonly diagnosed cancer in women as well as one of the leading causes of cancer-related death worldwide [1]. As the majority of breast cancers are hormone-dependent, inhibition of oestrogen receptor-α (ERα) signalling represents an effective therapeutic strategy [2]. While endocrine therapy is highly effective, a significant number of ERα-positive breast cancer patients will eventually progress and develop treatment-resistant, metastatic disease [3]. Prior research has shown that most of the resistant tumours remain ERα-positive and critically dependent on activity of this signalling axis, with activating *ESR1* gene mutations occurring in 15–40% of advanced cancers [4,5,6,7]. The amino acids within the loop connecting α-helices 11 and 12 (L536, Y537, and D538) of the ERα ligand binding domain are most frequently altered. Structural modelling studies suggested that once present, these *ESR1* mutations stabilize the agonist state of the receptor, thereby potentiating its activity in the absence of ligand [8,9]. The biological consequences of a constitutively active ERα, driving cancer cell proliferation and ERα target gene expression, have been functionally tested and confirmed using ectopic expression and CRISPR-Cas9-edited models, showing they share their genetic drivers with cells with the wildtype (WT) ERα [10,11,12,13,14,15]. Importantly, studies have shown that ERα mutations that occur in the ligand binding domain lead to reduced anti-oestrogen efficacy [16,17,18], while other treatment strategies, such as lasofoxifene and PROTACs, are still able to suppress the ERα signalling axis [19]. Therefore, there is an urgent need to discover novel effective therapies that can eradicate cancer cells with aberrant ERα as well as to understand the cellular response to these.

Herein, we study the response of an MCF7-derived, CRISPR-Cas9-generated cell line expressing ERα-Y537S to a large array of compounds. We show sensitivity to various clinically approved inhibitors, including CDK4/6 inhibitor ribociclib, which is a standard-of-care therapy in the treatment of metastatic ERα-positive breast cancer [20,21,22,23] and currently under evaluation in the neoadjuvant setting [24,25]. We demonstrate that ribociclib treatment leads to senescence in both WT and mutant ERα breast cancer models and induces broad-range drug tolerance to many chemotherapeutic agents clinically applied in the treatment of metastatic breast cancer. Strikingly, viability of cells undergoing ribociclib-induced cellular senescence is maintained via engagement of EGFR signalling, which may be therapeutically exploited in both WT and mutant ERα breast cancer.

## 2. Materials and Methods

### 2.1. Cell Lines

MCF-7 (Michigan Cancer Foundation-7) human breast carcinoma cell line was obtained from the Simak Ali laboratory (Division of Cancer, CRUK Labs, University of London Imperial College, London, UK). MCF-7 clone 1 (C1; referred to as ERα-MutA in this study) and MCF-7 clone 8 (C8; referred to as ERα-MutB in this study) mutant cell lines were generated using CRISRP-Cas9, as previously described [14]. Cell lines were maintained in regular Dulbecco’s modified Eagle’s medium (DMEM) supplied with 10% foetal bovine serum (FBS) and 1% penicillin/streptomycin and cultured at 37°C and with 5% CO_2_. All cell lines were authenticated and tested negative for mycoplasma contamination

### 2.2. Compounds

The following drugs were used in this study: Alpelisib (HY-15244), AZD-9496 (HY-12870), docetaxel (HY-B0011), doxorubicin (HY-10162), everolimus (HY-10218), fulvestrant (HY-13636), gefitinib (HY-50895), ipatasertib (HY-15186), NPS-2143 (HY-10007), olaparib (HY-10162), osimertinib (HY-15772), ribociclib (HY-15777), 4-Hydroxytamoxifen (HY-16950), and venetoclax (HY-15531). All of these drugs were purchased from MedChemExpress.

### 2.3. Viability Assay

Cells (500; 80 μL/well) were seeded into 384-well plates. On the following day, cells were treated with compounds using the HP D300 Digital Dispenser. The concentrations used were: 10, 5, 2.5, 1.25, 0.625, 0.312, 0.156, 0.078, 0.039, and 0.019 μM. At least three technical replicates were performed for each drug, and Phenylarsine oxide (PAO) and Dimethylsulfoxide (DMSO) were used as controls. Following six-day treatment, a CellTiter-Glo (CTG; Promega, Madison, WI, USA) assay was performed. Following the manufacturer’s instructions, cell viability was determined based on luminescent output detected using a Tecan microplate reader.

### 2.4. Protein Extraction and Immunoblotting

Cells were lysed in 2× Laemmli buffer (120mMTris, 20% glycerol, 4% SDS) supplemented with a protease and phosphatase inhibitor cocktail. Lysates were homogenized by sonication. Protein quantification was determined by bicinchoninic acid assay (23227, Thermo Fisher Scientific, Waltham, MA, USA) according to the manufacturer’s protocol. The measured values were normalized to a calibration curve and the protein concentration was calculated for each sample. DTT was added to protein lysates (30–50 μg), which were then heated at 95 °C for 5 min. Samples were loaded into Tris-Glycine gels next to the PageRuler Pre-Stained Protein Ladder (Thermo Fisher Scientific, Waltham, MA, USA) for determining protein size. Gels were run in SDS running buffer at 100 V. After protein separation according to the size by SDS-PAGE, protein samples were transferred onto nitrocellulose membranes using overnight wet-transfer method (0.09 A). Subsequently, the membranes were incubated in blocking solution (3% bovine serum albumin (BSA) in Tris-Buffered Saline with 0.1% Tween-20 (TBS-T) for at least 1 h. The membrane with the immobilized proteins were then incubated for 2 h or overnight with the appropriate primary antibody properly diluted in 3% BSA blocking buffer. The membranes were washed with TBS-T 1× three times and further probed with Odyssey secondary antibodies (anti-rabbit 800-nm channel and anti-mouse 680-nm channel) and signals detected using the Odyssey Imaging System.

### 2.5. Antibodies

Antibodies for HSP90 (sc-7947) and ERK1/2 (sc514302) were purchased from Santa Cruz Biotechnology. The ERα (MA5-14104) antibody was purchased from ThermoFisher. PARP (9542), EGFR (4267), Akt (40D4), p-Akt Ser473 (4060), ß-actin (4967), and p-ERK1/2 p44/42 (9102) antibodies were obtained from Cell Signalling Technology. Antibody against p-EGFR Y1068 (ab5644) was provided by Abcam.

### 2.6. Proteome and Phosphoproteome Analysis

Cells were cultured for 48 h with 600 nM ribociclib or left untreated. Cells were then collected using cold PBS and after centrifugation pellets were stored at −80 °C. Three biological replicates were performed. The mass spectrometry sample preparation and analysis was performed as previously described [26,27,28]. Differential phosphosites were identified by means of a *t*-test with multiple testing corrections using the Benjamini and Hochberg method. Kinase-substrate enrichment analysis was performed as previously described [29].

### 2.7. RNA Sequencing

Cells were pre-treated with 600 nM ribociclib or vehicle for 5 days in full medium containing 10% FBS. Total RNA was isolated in RLT buffer (Qiagen, Hilden, Germany) according to the manufacturer’s instructions and stored at −80 °C until analysis. Total RNA quality and quantity were evaluated by the 2100 Bioanalyzer using a Nano chip (Agilent, Santa Clara, CA, USA). Total RNA samples reporting an RNA integrity number (RIN) above 8 were subjected to library generation. The TruSeq Stranded mRNA sample preparation kit (Illumina, San Diego, CA, USA; RS-122-2101/2) was used to generate the strand-specific libraries as recommended by the manufacturer (Illumina, PArt #15031047 Rev. E). Twelve cycles of PCR were performed for every 3′ adenylated and adapter ligated cDNA fragments. Following that, the samples were processed and sequencing performed as previously described [30]. Counting and normalization of reads as well as differential gene expression analysis were performed using R package DESeq2 [31]. Gene-set enrichment analysis was executed according to standard instructions [32].

### 2.8. Senescence-Associated β-Galactosidase Staining

Cells were seeded into 6 well-plates, left to adhere overnight in standard conditions of 5% CO_2_ and 37 °C, and then treated with 600 nM ribociclib. Cells in media supplemented with 10% FCS were used as negative controls for senescence induction. After 5 days, cells were washed twice with PBS and fixed with 3.7% formaldehyde solution (in PBS). Subsequently, cells were incubated with X-gal staining solution (1 mg/mL X-gal, 40 mM citric acid/sodium phosphate buffer, 5 mM potassium ferricyanide, 5 mM potassium ferrocyanide, 2 mM MgCl2, 150 mM NaCl) at 37 °C, overnight. The following day, imaging was performed using a Zeiss Axiovert S100 inverted microscope (Zeiss, Jena, Germany).

### 2.9. Drug Screening

In order to evaluate possible synergistic effect with AZD-9496 or ribociclib in MCF-7 mutants cell lines (ERα-MutA and ERα-MutB), a drug screen was performed using the NKI compound collection of purchased drugs (Selleck GPCR, Kinase, Apoptosis, Phosphatase, Epigenetic, LOPAC, and NCI oncology). The library was stored and handled as recommended by the manufacturer. In the case of AZD-9496 treatment, cells were co-treated with the library compounds. For the ribociclib arm, cells harbouring mutated ERα were pre-treated for 5 days with the CDK4/6 inhibitor and then seeded at 80% confluency before adding the library compounds. Library compounds were diluted from the master plate in daughter plates containing complete DMEM medium, using the MICROLAB STAR liquid handling workstation (Hamilton Robotics, Reno, NV, USA). The diluted compounds were transferred from the daughter plates into 384-well assay plates, at a final concentration of 1 μM, in technical triplicates. Additionally, Phenylarsine oxide (1 µM) and DMSO (0.1%) were respectively used as positive and negative controls in each of the plates. After a six-day period, viability of cells was assessed by means of a CellTiter-Blue assay (G8081/2, Promega) according to the manufacturer’s manual. The data were transformed by means of a normalized percentage inhibition method. Computational analysis was performed in R.

## 3. Results

### 3.1. Drug Screening on ERα Mutant MCF-7 Cells

To characterize the anti-oestrogen response of MCF7-ERα-Y537S (ERα-MutA; generated using CRISPR-Cas9 as previously described [14]) and MCF7-ERα WT (ERα-WT) cells, we performed a viability assay in response to a logarithmic range of concentrations (0.01–10 μM) of two selective ERα degraders (SERDs), fulvestrant and AZD-9496. A significant decrease in sensitivity to ERα inhibitors was observed in the ERα-MutA model when compared to the ERα-WT, for both fulvestrant (Figure 1A) and AZD-9496 (Figure 1B), which is in agreement with previous in-vitro studies [14,17,33]. The treatment with 20 or 100 nM fulvestrant or AZD-9496 led to a reduction of ERα protein levels in ERα-WT models, while this was not observed in the ERα-MutA cell line (Figure 1C). These findings confirm previous reports [14,17,33] that demonstrated reduced affinity of SERDs for mutant ERα, resulting in decreased inhibition capacity. To identify compounds that reduce viability of breast cancer cells bearing a mutated ERα, we performed a drug screen in ERα-MutA cells using a composite NKI drug library containing 2277 drugs targeting various pathways (Figure 1D; composition of the drug library can be found in Appendix A). The ERα-MutA cells were treated with 1 μM of the screening compounds for six days, and viability was measured using CellTiter Blue (Figure 1D). These experiments identified numerous compounds that reduce the viability of ERα-MutA cells (Figure 1E and Appendix A; < 0.7 normalized viability), including ribociclib, palbociclib, docetaxel, alpelisib, everolimus, ipatasartib, methotrexate, and doxorubicin. To validate our results, the drug screen was performed in another independently CRISPR-Cas9-generated MCF-7 clone harbouring the *ESR1* Y537S mutation (ERα-MutB; generated as previously described [14]; Appendix A) and yielded comparable results (Pearson’s r^2^ = 0.9142; *p*-value < 0.0001). As independent validation, four hits from the screen (everolimus, alpelisib, ipatasartib, and doxorubicin) were selected and tested for their effect on viability on ERα-MutA and ERα-MutB cell lines across a concentration range, showing an inhibitory effect on cell proliferation for all of the compounds (Appendix A). As MCF-7 cells have a mutation in the *PIK3CA* gene, we explored if activity of compounds that target the PI3K pathway is affected by alterations of this gene by exploring the Genomics of Drug Sensitivity in Cancer database [34]. Overall, sensitivity to these compounds was comparable between cell lines harbouring either the *PIK3CA* mutant or WT allele, with an exception of alpelisib (*p*-value = 0.00007) and AZD6482 (*p*-value = 0.0455), which inhibited growth of *PIK3CA* mutant cell lines at lower concentrations than of the cells harbouring the WT allele (Appendix A).

In terms of their mechanism of action, drugs that target DNA-related processes (e.g., alkylating compounds, synthesis inhibitors), PI3K/Akt/mTOR pathway, and the cell cycle had the most pronounced effects on viability of ERα-MutA cells (Figure 1F). Cell-cycle-related drugs were the most potent inhibitors of viability in these cells, motivating us to focus further analyses on this drug class.

### 3.2. CDK4/6 Inhibitor Ribociclib Induces Senescence in ERα-Mut and ERα-WT Breast Cancer Models

To gain further insight into how particular compounds affect the cell cycle, we annotated the cell-cycle-related drugs based on their target class (Figure 2A), showing that drugs that target cyclin-dependent kinases and tubulin decrease the viability of ERα-MutA cells. Specifically, both ribociclib and docetaxel, clinically used for treatment of ERα-positive breast cancer, exhibited an anti-proliferative action on the ERα-MutA cells (Figure 2A). As CDK4/6 inhibitors represent standard-of-care therapeutics for patients with ERα-positive metastatic disease who relapsed after prior endocrine therapeutics [35], the phase of the disease in which ERα mutations are most apparent [36], we focused on ribociclib for downstream analyses on ERα-WT and ERα-MutA. Cell viability assays using a range of concentrations (0.01–10 μM) illustrated that treatment with this CDK4/6 inhibitor reduced viability to a similar extent in both models (Figure 2B). In agreement with previous studies [37,38,39], ribociclib treatment did not induce apoptosis in our models, as no PARP cleavage was observed by western blot analysis of ERα-WT cells (Appendix A). We further investigated the cellular response to ribociclib and performed RNA sequencing following 48-h exposure to the drug. Ribociclib induced a significant downregulation of various cell-cycle genes (e.g., *FANC* and *CDCA7*) as well as upregulation of genes involved in senescence (e.g., *CEACAM11* and *SEPRINA11*), both in ERα-MutA (Figure 2C) and ERα-WT models (Appendix A). Furthermore, over-representation analysis of genes upregulated by ribociclib treatment revealed that these genes are involved in cellular processes, such as phagocytosis, adhesion, as well as senescence (Figure 2D). Conversely, over-representation analysis of the downregulated genes showed that ribociclib led to decrease in expression of genes related to the cell cycle, DNA replication, and steroid hormone response (Appendix A), confirming a previous multi-omics study [40].

To validate the observed increased senescence and decreased cell-cycle-related signalling, we next performed a whole-proteome mass spectrometry analysis of cells treated with ribociclib or vehicle for 72 h. Gene-set enrichment analysis showed that ribociclib treatment led to enrichment of a senescence signature in both ERα-WT and ERα-MutA cell line models (nominal *p*-value: ERα-WT = 0.021 and ERα-MutA = 0.004) and significant downregulation of cell-cycle-related E2F targets on protein level (nominal *p*-value: ERα-WT = 0.001 and ERα-MutA = 0.001; Figure 2E). To validate the RNA sequencing and proteomics observations related to senescence induction in ERα-WT and ERα-MutA models upon ribociclib treatment, β-galactosidase activity staining was performed. We observed an increase in number of X-gal-positive cells following five-day ribociclib treatment of both ERα-WT and ERα-MutA models, confirming entry to senescence (Figure 2F,G).

### 3.3. CDK4/6 Inhibition Induces Broad-Spectrum Drug Resistance and EGFR Dependence

As senescence has been previously shown to alter cancer progression as well as therapy response [41], we sought to identify vulnerabilities of CDK4/6 inhibitor-induced senescent cells. Therefore, we performed a drug screen with a composite NKI drug library (Appendix A) according to the procedure summarized in Figure 3A. In brief, after five-day treatment with either vehicle or ribociclib (600 nM), cells were seeded into well plates, and the respective treatment was continued. Subsequently, compound library drugs were added (1 μM) and cells cultured for six days. Following six-day exposure to the library drugs, cell viability was assessed using CellTiter Blue, after which the ribociclib arm was compared to the vehicle control. Interestingly, we observed that ribociclib treatment led to reduction in sensitivity to a number of chemotherapeutics (Figure 3B; −log_10_(q-value) < 1, and viability difference < −0.3), including abemaciclib and gemcitabine, which are used for treatment of breast cancer. These data suggest that induction of senescence through ribociclib is accompanied by a decreased responsiveness to various compounds, analogous to our previous observations on glucocorticoid receptor-mediated growth arrest induction in lung cancer [30]. This decreased effectiveness of various anti-cancer drugs following ribociclib was confirmed by performing a drug screen with the same set-up in ERα-MutB cell line (Appendix A).

Next to decreased efficacy of numerous therapeutics upon ribociclib treatment, we observed a significant increase in sensitivity to 18 inhibitors (including IPI-549, bromosporine, BWB70C; Appendix A), two of which are EGFR inhibitors (osimertinib and AST-1306) (Figure 3B). Increased sensitivity to these two EGFR inhibitors was confirmed ERα-MutB cells (Appendix A).

### 3.4. CDK4/6 Inhibitors Increase EGFR Pathway Activity

To study changes in the EGFR pathway upon CDK4/6 inhibition, we next inspected EGFR signalling both at the mRNA and protein level using RNA sequencing and whole-proteome mass spectrometry analysis. Treatment with ribociclib led to a significant increase in EGFR signalling mRNA-based activity (signalling by EGFR, Reactome, R-HSA-177929) in both ERα-WT and ERα-MutA cell lines on RNA level (Figure 4A). The latter was accompanied by a significant receptor tyrosine kinase (signalling by receptor tyrosine kinases, Reactome, R-HSA-9006934) pathway enrichment at the protein level, as demonstrated by gene-set enrichment analysis of the whole-proteome datasets (Figure 4B).

Furthermore, to investigate whether ribociclib treatment leads to an increase in EGFR pathway activity, we performed phospho-proteomics and subsequent Kinase Substrate Enrichment Analysis (KSEA) [29]. A significant (*p*-value < 0.05) increase in activity of AKT2, PRKAA1, MAPK1, AKT3, AKT1, and MAPK8 was observed following ribociclib treatment in ERα-MutA cell line (Figure 4C). In addition, ribociclib-treatment resulted in reduction of CDK1, AURKA, WEE1, and CDK2 activity (Figure 4C), which is concordant with its CDK-inhibitory role [40,42]. The same observation was made in ERα-WT MCF-7 cells following treatment with ribociclib (Appendix A). These findings were confirmed by western blot analyses, demonstrating increased phosphorylation of both EGFR and ERK1/2 upon ribociclib treatment, both in ERα-WT and ERα-MutA models (Appendix A). Our discoveries are further substantiated by the data of the FELINE trial, in which single-cell RNA sequencing was performed on breast cancer samples from patients undergoing treatment with letrozole or ribociclib [43]. Inspection of the single-sample gene-set enrichment data revealed that ribociclib selectively alters the cell cycle gene networks as well as the EGFR- and PI3K-related pathways (Appendix A).

To show that cancer cell viability in the ribociclib-induced growth arrested state is dependent on the EGFR pathway, we tested whether inhibitors of the EGFR pathway (other than Osimertinib and Gefetinib) affect viability of ERα-Mut cells treated either with vehicle, the SERD AZD-9496 (20 nM), or ribociclib (600 nM). Increased sensitivity to EGFR inhibitors was observed under ribociclib treatment (Figure 4D; ERα-MutA *p*-value = 0.0237; ERα-MutB *p*-value < 0.0001) but not in the AZD-9496-treated cells (Figure 4D; ERα-MutA *p*-value > 0.9999; ERα-MutB *p*-value < 0.5239), suggesting that the observation is specific for ribociclib-treated cells.

## 4. Discussion

Endocrine resistance constitutes a major clinical problem in breast cancer treatment, and tremendous efforts have been made to uncover crucial mechanisms that underlie this phenomenon [44]. Breast cancer cells harbouring *ESR1* alterations in the region encoding the ligand binding domain have been described in 15–40% of ERα-positive metastatic tumours, while they are rarely present in primary tumours [4,5,6,7]. In the preclinical setting, various cellular models of endocrine resistance have been developed with the purpose of identifying mechanisms that sub-serve the mutant *ESR1* action [10,11,12,13,14,15]. In this study, we investigated the drug response of ERα-mutant cells, which are resistant to ERα degraders [14,17], by performing a compound screen with a library that contains 2277 compounds targeting diverse cellular signalling pathways. The viability of Y537S ERα-mutant cells was particularly sensitive to cell cycle inhibitors, including ribociclib, a selective, reversible CDK4/6 inhibitor. Furthermore, we evaluated whether there are any differences in response of ERα-mutant and ERα-WT MCF-7 cell lines to ribociclib. Our results are consistent with clinical studies proving similar inhibitory activity of CDK4/6 inhibitors regardless of the presence of *ESR1* mutations [45,46]. Treatment with CDK4/6 inhibitors has been shown to significantly improve progression-free survival in metastatic ERα-positive breast cancers when prescribed in combination with either letrozole as first-line therapy in endocrine-sensitive disease [20,22,23] or with fulvestrant as second-line therapy in endocrine-resistant tumours [21]. While ribociclib treatment represents a major improvement in clinical interventions, not all patients benefit and all patients with metastatic disease inevitably experience disease progression [47,48,49]. One of the pressing clinical questions is whether an optimal combination therapy in conjunction with CDK4/6 inhibition exists and if that will benefit the majority of breast cancer patients [50].

Molecular profiling using RNA sequencing and proteomics revealed that ribociclib-treatment triggers a significant reduction in expression of cell cycle-related genes (e.g., E2F targets; as previously reported for other CDK4/6 inhibitors [51]) and an increase in senescence markers, which was also confirmed by β-galactosidase activity assays and is in line with prior studies [37]. In terms of cancer treatment, it has been demonstrated that cancer cells that reside in a prolonged senescent state may metastasize and is preceded by an asymptomatic period where the tumour is present but does not progress [52]. Breast cancer is particularly prone to this phenomenon, where dormant tumour cells persist, and disease progression may initially not be clinically apparent [53]. In this dormant state, appropriate stimuli, such as growth factors and hormones, can determine the cell fate and push the senescent tumour cells towards cell cycle re-entry and consequently cell proliferation [54,55]. We performed a drug screen aimed at understanding the pathway dependencies in cancer cells with ribociclib-induced senescence. Interestingly, we observed that CDK4/6 inhibition diminished the impact on viability of many cancer drugs. Consistent with our findings, several studies have shown that co-treatment with CDK4/6 inhibitors reduced the therapeutic effect of a number of chemotherapeutic compounds, such as doxorubicin, gemcitabine, methotrexate, cisplatin, etoposide, and taxanes [56,57,58], some of which were also less effective in our compound screen. The observation that CDK4/6 inhibition may give rise to drug tolerance warrants additional studies to assess whether the timing of drug administration can ameliorate this effect of ribociclib, as has been proposed for palbociclib in pancreatic cancer [56].

In terms of pathway dependences, we observed that CDK4/6 inhibitor-induced senescent cells are vulnerable to EGFR pathway inhibitors. Using transcriptomic and proteomic techniques, we demonstrated that ribociclib treatment potentiated the EGFR signalling axis, ultimately leading to a significant increase in the activities of downstream kinases, particularly PRKAA1, AKT1, and MAPK1. In addition, by means of western blot analysis, we observed an increase in phosphorylation levels of EGFR and its downstream targets ERK1/2, confirming the activation of the EGFR pathway following ribociclib treatment. Activation of receptor tyrosine kinase signalling in growth arrested cancer cells has been shown to be important for maintaining tumour cell viability under lethal drug-dose exposure [59], nutrient depletion [60], and glucocorticoid-treatment [30]. Importantly, it was previously shown that breast cancer cells resistant to CDK4/6 inhibitors can be successfully eradicated using drugs that target the downstream signalling of the receptor tyrosine kinases, such as lucitanib (FGFR) and alpelisib (p110α-selective inhibitors of PI3K) [61,62,63]. In conjunction with the latter and research on oesophageal tumours [64] and non-small cell lung cancer [65], our data suggest that activation of receptor tyrosine kinase signalling and acquired inhibitor vulnerability may be an early adaptation mechanism of response to CDK4/6 inhibitors, which could persist in fully resistant cancers but opens up the possibility of targeting CDK4/6 inhibitor resistance with inhibitors of receptor tyrosine kinase signalling cascades. Lastly, further studies are needed to understand the biological underpinnings of EGFR activation following ribociclib-induced senescence. It was reported that CDK4/6 inhibitors can be trapped in the lysosomes, subsequently influencing key regulatory cellular processes, such as protein turnover [66]. Moreover, EGFR is degraded in the lysosome [67], and resistance to EGFR inhibitors may arise due to impartment of its effective trafficking [68]. Thus, we hypothesize that the increase in EGFR pathway activation may be a consequence of altered lysosomal trafficking due to trapping of ribociclib in the lysosomes. Addressing this hypothesis experimentally would be of high interest and would further shed light on the effects of CDK4/6-targeted therapy in breast cancer.

## 5. Conclusions

Identifying novel therapeutic targets based on cancer vulnerabilities remains a promising strategy to effectively delay disease progression and improve outcome of ERα-positive breast cancer patients regardless of the presence of *ESR1* mutations. We observed that ribociclib-induced senescent breast cancer cells are tolerant to a large number of compounds, with viability being maintained by EGFR activation. While our study benchmarks a large number of compounds and their effect on mutant cells in vitro, our findings may have been influenced by the choice of screening read-out assay (an ATP-based measurement) as well as the concentration of screening compounds used. Therefore, alternative screen set-ups may yield additional targets that may be taken further for preclinical evaluation.

Additional preclinical models, including ex-vivo tumour cultures and patient-derived xenografts, evaluating the reduction of drug sensitivity following CDK4/6 inhibitor treatment as well as synergistic potential of CDK4/6 and EGFR inhibitors in CDK4/6 inhibitor-naïve models are required to further confirm our observations. Our results open new avenues in exploration of drug sensitivity of cells expressing WT and aberrant ERα and suggest that optimisation of therapy sequencing may improve effectiveness and ultimately lead to reduced tumour burden. Overall, our study highlights a wide-spread reduction in sensitivity to anti-cancer drugs accompanied with an acquired vulnerability to EGFR inhibitors following CDK4/6 inhibitor treatment.

## Figures and Tables

**Figure 1 cancers-13-06314-f001:**
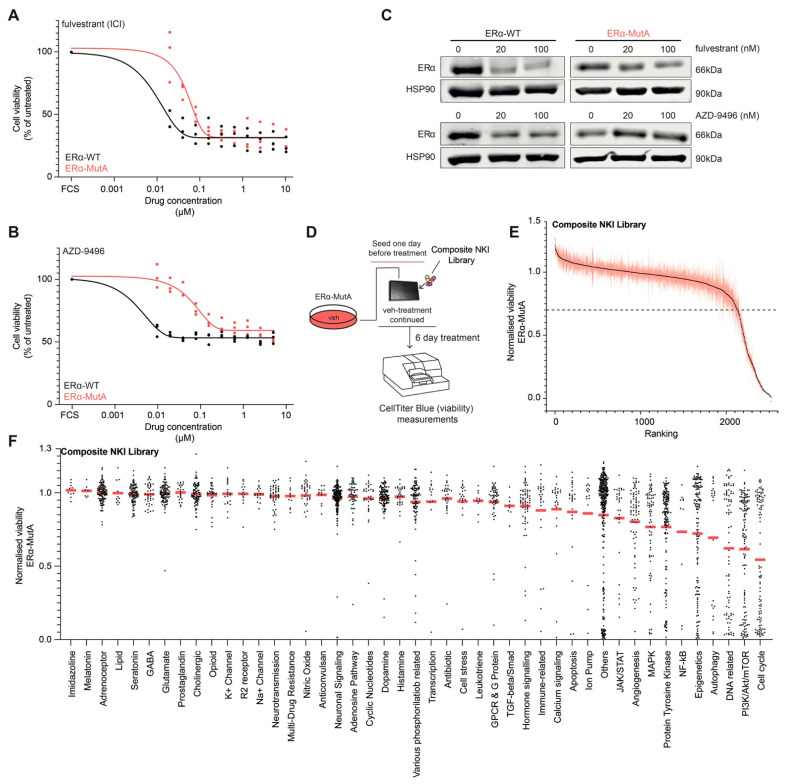
ERα mutant cells are sensitive to various inhibitors of the cell cycle. (**A**) ERα-MutA and ERα-WT cells were cultured with ER degrader fulvestrant. After six days, cell viability was measured using CellTiter-Glo. Full circles depicted independent replicate (*n* = 3) values, while the lines represent the curve fitting for sigmoidal (4PL) model (black, ERα-WT; red, ERα-MutA). (**B**) ERα-MutA and ERα-WT cells were cultured with ERα degrader AZD-9496. After six days, cell viability was measured using CellTiter-Glo. Full circles depicted independent replicate (*n* = 3) values, while the lines represent the curve fitting for sigmoidal (4PL) model (black, ERα-WT; red, ERα-MutA). (**C**) Representative western blot showing expression of ERα in ERα-MutA and ERα-WT cell lines treated with 20 nM and 100 nM fulvestrant or AZD-9496 for 24 and 48 h, respectively. Hsp90 was used as loading control (*n* = 3). (**D**) Schematic representation of the drug screen: ERα-MutA cells were seeded and cultured in full medium. On the following day, the drugs from the library were added at the concentration of 1 μM. After six-day treatment, CellTiter Blue assay was performed and viability measured using fluorescence reader. (**E**) Ranked plot showing viability of ERα-MutA cells following a six-day with 1 μM of the library compounds. Drugs that reduce the viability below 0.7 are considered effective. Black dots and red shade around these represent the mean viability and SD per compound, respectively (*n* = 3). (**F**) Normalized viability per compound category of the drug screen in ERα-MutA cells. Red line represents the mean viability per category, and black dots represent viability value for each of the compounds (*n* = 3). Appendix A contain the western blot original material including uncropped blot images and densitometry readings/intensity ratio of western blot bands, respectively.

**Figure 2 cancers-13-06314-f002:**
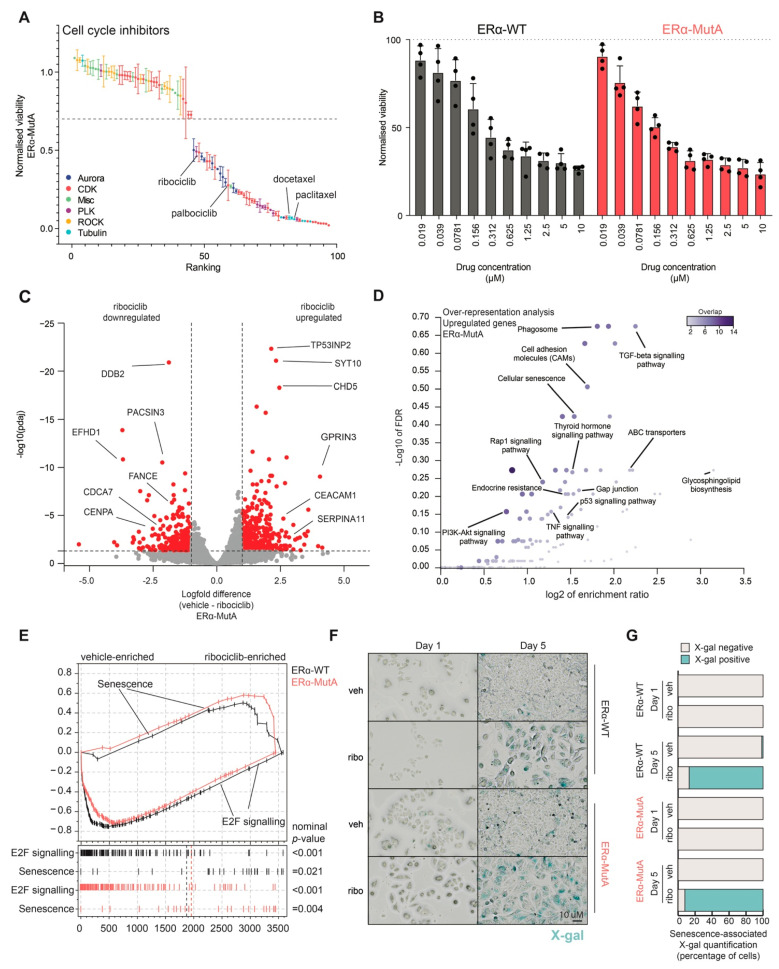
CDK4/6 inhibitors induce senescence in cells harbouring mutant ERα. (**A**) Compounds belonging to the cell cycle category (dark blue, aurora; red, CDK; green, misc.; purple, PLK; orange, ROCK; light blue, tubulin) were annotated based on their target and ranked according to their effect on EαR-MutA viability. Mean ± SD are depicted (*n* = 3). (**B**) ERα-MutA and ERα-WT cells were cultured with CDK4/6 inhibitor ribociclib. After six days, cell viability was measured using CellTiter-Glo. Bars depict mean value ± SD (*n* = 4). (**C**) Volcano plot depicting log2fold differences from an RNA sequencing experiment of ERα-MutA cells treated with either Vehicle or 600 nM ribociclib for five days (*n* = 2). Differentially expressed genes (log2fold < −1 and >1; p-adj = 0.05) are shown in red. Adjusted *p*-values (padj) were determined by DESeq2 (Wald test *p*-values corrected for multiple testing using Benjamini and Hochberg method). (**D**) Over-representation analysis of genes upregulated by ribociclib treatment in ERα-MutA cells (*n* = 2). FDR was computed using WebGestalt tool. (**E**) Gene-set enrichment profile of whole-proteome data for E2F signalling and senescence gene-set for ERα-MutA and ERα-WT cell lines (*n* = 4). Nominal *p*-values were determined by GSEA software. (**F**) Representative images of senescence-associated β-galactosidase (X-gal) stained cells, untreated (veh), or ribociclib treated (ribo) (*n* = 3). Scale bar, 10 μm. (**G**) Quantification of at least 200 cells from senescence-associated β-galactosidase experiments represented in fraction of negative (grey) and positive cells (blue) (*n* = 3).

**Figure 3 cancers-13-06314-f003:**
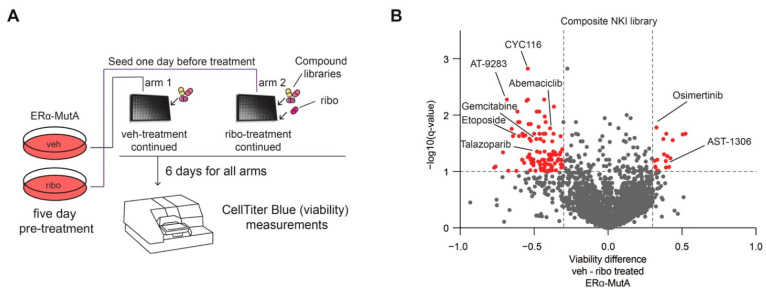
High-throughput drug screen identifies EGFR as a vulnerability of CDK4/6 inhibitor treated cells. (**A**) Schematic representation of the drug screen: ERα-MutA cells were seeded. On the following day, the drugs from the library were added at the concentration of 1 μM. After six-day treatment. CellTiter Blue assay was performed and viability measured using fluorescence reader. (**B**) Volcano plot depicting the viability differences between two arms of the screen—vehicle or Ribociclib pre-treated ERα-MutA cells. Significant hits are depicted in red, and are based on two-tailed unpaired *t*-test with Welch corrections (with multiple correction Benjamin, Krieger, and Yekutieli test). Compounds without a differential effect are depicted in grey.

**Figure 4 cancers-13-06314-f004:**
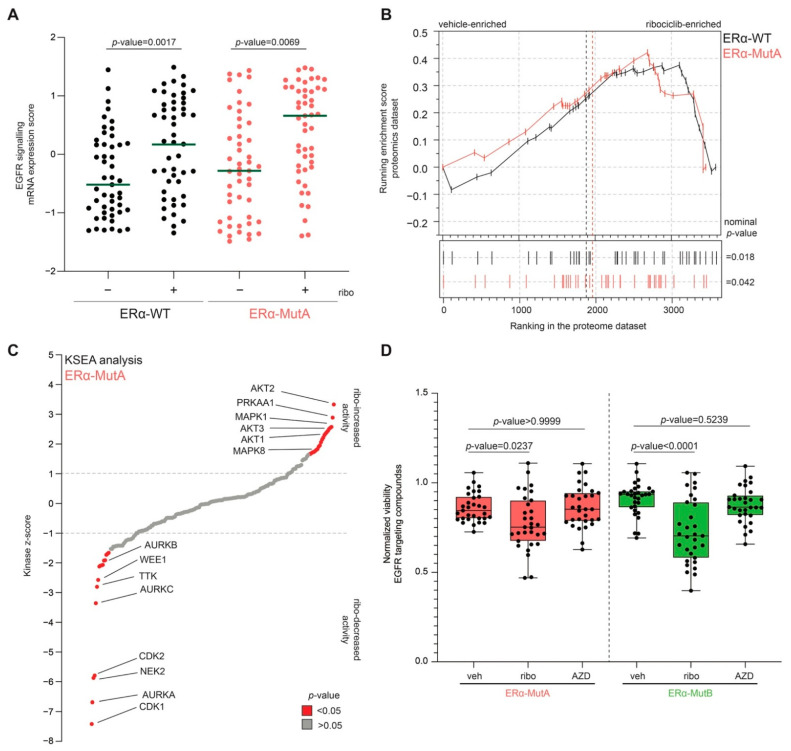
Ribociclib increases the activity of EGFR signalling pathway. (**A**) Scatter plot depicting expression of genes involved in EGFR signalling pathway. Each dot represents a gene, and the green line represents the mean value per condition (*n* = 2). *p*-Values were determined by a two-tailed Wilcoxon matched-pairs signed rank test. (**B**) Gene-set enrichment profiles of whole-proteome data of ERα-MutA and ERα-WT for signalling by receptor tyrosine kinases (M27870) (*n* = 4). (**C**) Plot depicting kinases ranked based on the kinase activity z-score calculated using kinase-substrate-enrichment-analysis of phospho-proteomic data (*n* = 4). The significant kinases are marked red (*p*-value < 0.05, determined by KSEA software). (**D**) Normalized viability of ERα-MutA and ERα-MutB following a five-day treatment with vehicle (veh), ribociclib (ribo), or AZD-9496 (AZD) in combination with 32 different EGFR inhibitors. *p*-Values were determined using Dunn’s multiple comparison test.

## Data Availability

All genomic and mass spectrometry data generated in this study have been deposited in the Gene Expression Omnibus (GEO) and Proteomics Identification (PRIDE) databases, under accession numbers GSE182288 and PXD02809, respectively. The remaining data are available within the Article, Appendix A, or available from the authors upon request.

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
