# Peer review of "Ribociclib Induces Broad Chemotherapy Resistance and EGFR Dependency in ESR1 Wildtype and Mutant Breast Cancer"

_cancers, 2021, doi:10.3390/cancers13246314_

Round 1

Reviewer 1 Report

Generally a well thought out study and well presented. A few typos throughout (including omission of small words that improve the fluency) including in supplementary figure legends.

Overall the study would be enhanced by analysis of patient samples, particularly from those who had been treated by the drugs involved to, assess presence of senescent cells and/or explant culture showing increased/decreased sensitivity to the drugs following Ridociclib treatment.

Some minor comments

In section 2.6 you state differential proteins were determined using t-test, what correction was done for multiple comparisons?

The methods need to include information on GSEA, KSEA etc. analysis

Results

3.1 SERDs needs defining

The reference to agreement with previous findings may be better placed in the discussion (however, I am also OK with it in the results as it is fundamental to demonstrating that the treatment worked as expected and as such the resulting data is likely to be valid)

Is ’metotraxate’ supposed to be ‘methotrexate’? I can only find methotrexate in the supplementary table.

Discussion

In the second paragraph the word ‘faith’ should probably be ‘fate’

Discussion of EGFR activation should be stated more clearly, bringing in the supplementary data showing increased phosphorylation of ERK1/2 as confirmation of the key proteins mentioned in the discussion.

Reviewer 2 Report

Summary:

This study describes that senescence induced by ribocliclib may lead resistant in ER+ breast cancer cells to chemotherapy.  Additionally, the authors propose EGFR pathway activation as potential mechanism to maintain the senescence profile. To that end, the authors generated potent tools, MCF7 and MCF7-ESR1 (Y537S) mutant breast cancer cells lines to perform pharmacological screening using a large number of drugs to identify 1.  drugs which can overcome the resistant mechanism associated to ESR1 (Y537S) mutation and 2. drugs which can bypass the senescence mechanism associated to ribociclib. 1. Indeed, the authors show by sequencing and proteomic analysis the downregulation of genes associated to inhibition of cell cycle progression as well as increasing of genes involved in senescence phenotype, which were validated through orthogonal systems such as IHC or western blot. At the end, the authors made a POC to sensitive the cells treated with ribociclib with combination of EGFRi that results in a greater cell viability inhibition.

Major comments:

  1. Cell titer glo is not a good readout for compounds promoting senescence, since is based on ATP levels, which can be altered by the size of the cells or by the metabolism. Author may consider monitoring cell counting (PI) for some findings to confirm data generated by CTG.
  2. Authors used AZD9496 as a tool to validate the model MCF7-ERalpha Y537S, however, this compound has been tested in PDX models harboring this mutation, resulting in sensitivity, it is worth the author have a look and clarify the differences. Lei J et al., 2019 J Cancer Metastasis Treat. 2019 ; 5.
  3. Authors identify from the screening a subset of compounds that reduces the cell viability of ER mutant cells lines however, we do not know whether those are specific of mutant since there is not a head-to-head comparison versus MCF7 wt. So, I do not see what the value is to make the screening in mutant if that are not being comparable with wt cells, I would appreciate if the authors can clarify this point.
  4. Moreover, MCF7 is a cell line with mutation in PI3K (E545K), so, it is quite normal they respond to alpelisib, everolimus, ipatasertib and others responsible for PI3K signaling pathway inhibition. It will be interesting to validate those findings in different cell line, maybe PI3K wt.
  5. Additionally, the authors maybe would like to elaborate more on mechanisms of sensitivity or resistant associated to senescence mediated by ribociclib. Especially on EGFR hypothesis to make clear the statement on the role of EGFR plays to maintain senescence phenotype. The author could make a bgal assay in the presence of EGFRi to understand whether the senescence has been reduced. Do anti-senolytics compounds have been included in the library tested to validate the hypothesis

Minor comments:

On figure 3B, at the volcano graph, there is mistake, AZD-1306 is pointed out, but my guess is AST-1306.

On supplemental figure 4B, p-EGFR differences are not clear. Maybe by quantification of p-protein levels can let us know whether differences between treated and untreated are significant.

Overview

Overall, the manuscript supports the hypothesis on senescence induced by ribociclib is the root of chemotherapy resistance and to propose sequential treatments to overcome the senescence phenotype, however to make stronger those hypotheses, I think more mechanistic assays should be performed. I found out quite interesting work, but too preliminary for the statements shared.

Reviewer 3 Report

This research article is clear, comprehensive, well-structured and relevant to the field.

References are current and there are only 2 of 66 self-citations. 

The design of the experiments are adecuate and the methods are detailed and well-described; results are reproducible.

Figures and tables show the data properly and in an appropriate manner, they are easy to read, interpret and understand.

Statements and conclusions are coherent and supported by the listed citations.

Therefore, in my opinion this paper does no need further changes.
